# Evaluation of the Diagnostic Accuracy of Nasal Cavity and Nasopharyngeal Swab Specimens for SARS-CoV-2 Detection via Rapid Antigen Test According to Specimen Collection Timing and Viral Load

**DOI:** 10.3390/diagnostics12030710

**Published:** 2022-03-14

**Authors:** Seungjun Lee, Kristin Widyasari, Hye-Ryun Yang, Jieun Jang, Taejoon Kang, Sunjoo Kim

**Affiliations:** 1Department of Laboratory Medicine, Gyeongsang National University Changwon Hospital, Changwon 51472, Korea; sjlee0318@gmail.com (S.L.); kristinwidyasari@gmail.com (K.W.); 2Gyeongnam Center for Infectious Disease Control and Prevention, Changwon 51154, Korea; fsglg@hanmail.net (H.-R.Y.); jjegncdc@gmail.com (J.J.); 3Bionanotechnology Research Center, Korea Research Institute of Bioscience and Biotechnology (KRIBB), Daejeon 34141, Korea; kangtaejoon@kribb.re.kr; 4Gyeongsang National University College of Medicine, Gyeongsang Institute of Health Sciences, Jinju 52727, Korea

**Keywords:** SARS-CoV-2, COVID-19 testing, rapid antigen test, nasal cavity, detection

## Abstract

The rapid diagnosis of SARS-CoV-2 is an essential aspect in the detection and control of the spread of COVID-19. We evaluated the accuracy of the rapid antigen test (RAT) using samples from the nasal cavity and nasopharynx based on sample collection timing and viral load. We enrolled 175 patients, of which 71 patients and 104 patients had tested positive and negative, respectively, based on real time-PCR. Nasal cavity and nasopharyngeal swab samples were tested using STANDARD Q COVID-19 Ag tests (Q Ag, SD Biosensor, Korea). The sensitivity of the Q Ag test was 77.5% (95% confidence interval [CI], 67.8–87.2%) for the nasal cavity and 81.7% (95% [CI, 72.7–90.7%) for the nasopharyngeal specimens. The RAT results showed a substantial agreement between the nasal cavity and nasopharyngeal specimens (Cohen’s kappa index = 0.78). The sensitivity of the RAT for nasal cavity specimens exceeded 89% for <5 days after symptom onset (DSO) and 86% for Ct of *E* and *RdRp* < 25. The Q Ag test performed fairly well, especially in the early DSO when a high viral load was present, and the nasal cavity swab can be considered an alternative site for the rapid diagnosis of COVID-19.

## 1. Introduction

Since the initial outbreak in 2019, SARS-CoV-2 has caused the historic COVID-19 pandemic and remains an ongoing serious public health concern worldwide. The starting point for managing the spread of COVID-19 is the rapid and accurate detection of SARS-CoV-2. The laboratory diagnosis of SARS-CoV-2 is essential for detecting infected patients and beginning the process of identification, isolation, and tracing to prevent the spread of the virus within the community. Although the molecular diagnostic method features very high sensitivity and is regarded as the gold standard of detection, it has some drawbacks: carryover contamination, the necessity of well-trained technicians, expensive equipment, and difficulty of implementation in resource-poor countries [1,2]. In particular, it takes several hours or even a few days to obtain a result from a molecular test, which hampers immediate response to positive cases [3].

To overcome the limitations of the molecular diagnosis of SARS-CoV-2, multiple diagnostic methods have been developed and studied, including the rapid antigen test (RAT) [4,5]. An antigen test is an immunoassay that detects the presence of a specific viral antigen, which implies current viral infection. The RAT detects viral antigen using immobilized coated antibodies specific to SARS-CoV-2 on the device, which react with samples that are contaminated with SARS-CoV-2 fragment protein (antigen) [6]. RATs are relatively inexpensive, can be used at the point-of-care, and are time-effective. Most of the currently authorized tests return results in approximately 15–30 min [7,8]. Even now, antigen tests are commonly used in the diagnosis of other respiratory pathogens, including influenza virus and respiratory syncytial virus [9]. Due to its importance for reducing the burden on molecular diagnostic laboratories, the World Health Organization (WHO) suggested the potential use of triage tests to rapidly identify SARS-CoV-2 patients to avoid the cost of high-priced molecular testing and suggested the potential utility of RATs for diagnosis [10,11]. RATs could be used as effective screening tools in crowded high-risk areas, such as airport immigration checkpoints, nursing homes, and cultural performance halls [3].

Furthermore, early detection of SARS-CoV-2 is crucial in the triage of patients with acute/subacute diseases during the COVID-19 pandemic. Therefore, the application of RATs as a rapid detection platform may give an early and rapid diagnosis of COVID-19, especially for asymptomatic patients with high-risk acute diseases. Early diagnosis of COVID-19 helps give the best clinical judgment before delivering healthcare, and at the same time minimize cross-infection of COVID-19 among patients, protect the healthcare workers from the possibility of cross-infection and maintain the service in the healthcare center [12,13]. To give an accurate result in detection of SARS-CoV-2, it is essential to find a safe and reliable diagnostic specimen type. SARS-CoV-2 spreads via respiratory droplets from an infected person, which is propelled through the air and deposited on the mucous membranes of the mouth, nose, or eyes of persons who are nearby [14]. Given this fact, nasopharyngeal or oropharyngeal swab specimens are generally considered the primary choice as specimens for detection of SARS-CoV-2 and other respiratory viruses [15,16]. However, as naso- and oropharyngeal sampling are uncomfortable and painful, some patients tend to avoid SARS-CoV-2 testing. In addition, a well-trained expert is required for the accurate and safe sampling from this area. Therefore, an alternative specimen sampling site that gives results as good as those from naso- or oropharyngeal specimens and does not cause pain for patients is desirable as a first step in the diagnosis of COVID-19.

Nasal cavity swabbing refers to the swab sampling of the anterior nasal cavity, which is bounded anatomically by the nares anteriorly and the hard–soft palate transition posteriorly [17]. A nasal cavity swab can be used to collect specimens without extreme pain and the necessity of experts for sampling [18]. Given that in previous studies, nasal cavity swab specimens have been reported to be as effective as nasopharyngeal swabs for identifying influenza and respiratory syncytial virus (RSV), while causing significantly less pain [19,20], we consider the nasal cavity swab to be considerably practical as an alternative method of specimen collection for the detection of SARS-CoV-2. In this study, we evaluated the analytical performance of RATs for the detection of SARS-CoV-2 from nasopharyngeal and nasal swab specimens. We hypothesized that rapid antigen testing has potential diagnostic value for SARS-CoV-2 detection and thus can be useful in resource-poor countries with high SARS-CoV-2 prevalence or in special circumstances of close contact with confirmed cases. Therefore, we prospectively conducted immunodiagnostic tests with a RAT assay using specimens from two different sampling locations: nasopharynx and nasal cavity. Furthermore, we also analyzed the data according to days from symptom onset to sampling collection (DSO) and the cycle threshold (Ct) values of real-time (RT)-PCR. We also assumed that the color intensity of the RAT may be correlated with viral numbers; therefore, we compared Ct values for RT-PCR with the color intensity of the RAT.

## 2. Materials and Methods

### 2.1. Subjects

This study was conducted with 71 patients confirmed by RT-PCR to be infected with SARS-CoV-2 and 104 uninfected controls. A SARS-CoV-2-positive case was defined as any patient with a cycle threshold (Ct) value less than 35. Each SARS-CoV-2-infected patient was admitted to a hospital or institute designated for COVID-19 treatment between December 2020 and January 2021. These patients were either asymptomatic or mildly symptomatic. Uninfected controls were selected from patients who were admitted to Gyeongsang National University Changwon Hospital (GNUCH) for the management of other diseases during the same period and were screened with RT-PCR before admission. All participants agreed to this study and submitted their written informed consent. SARS-CoV-2-positive patients submitted written consent after discharge.

### 2.2. Sample Collection

Nasal cavity swabbing was carried out by inserting a flocked swab (NFS, Noble Biosciences, Hwaseung, Korea) into the nostril after blowing the nose to a depth of 1 to 2 cm and rotating three times against the surface of the nasal cavity. Following sample collection, the swab was put into a sterile conical tube. The nasopharyngeal specimen was taken from the same nostril to a depth of 5 to 7 cm using a thin, narrow-shaped flocked swab (NFS-1, Noble Biosciences). These two samples from the nasal cavity and the nasopharynx were used for the RAT. Another nasopharyngeal specimen was taken from the other nostril for molecular detection by RT-PCR. This swab was inserted into a CTM tube (Clinical Transport Medium, Noble Biosciences) with 2 mL of the virus transport medium (VTM). All specimens were transferred to GNUCH within 2–6 h and stored at room temperature.

### 2.3. The Rapid Antigen Test (RAT)

The STANDARD Q COVID-19 Ag test (SD Biosensor, Suwon, Korea), a rapid chromatographic immunoassay for the qualitative detection of specific SARS-CoV-2 nucleocapsid antigens, was used to detect SARS-CoV-2. The STANDARD Q COVID-19 Ag test, hereinafter referred as the Q Ag test, was performed following the manufacturer’s instructions. Briefly, each swab was inserted into a buffer tube and mixed 10 times. After squeezing the swab to the tube wall, three drops of the reaction mixture were applied to the device. After 15 min, the result was interpreted. The subjective color intensity score was recorded within a range of 0 (negative) to 21 (very strong) (Figure 1).

### 2.4. Real-Time Reverse-Transcription Polymerase Chain Reaction (RT-PCR)

A STANDARD M nCoV Real-Time Detection kit (SD Biosensor) was used for detection by RT-PCR assay of SARS-CoV-2 nucleic acids present in the specimens. The RT-PCR assay targets 2 highly conserved genes, i.e., envelope gene (*E*) and RNA-dependent RNA polymerase gene (*RdRp*) of SARS-CoV-2. The result was interpreted as positive only if the Ct values of both target genes were within the cutoff (≤35), and negative if they were outside the cutoff or if there was no amplification.

### 2.5. Statistical Analysis

Diagnostic performance measures of the Q Ag test, including sensitivity, specificity, positive predictive value, negative predictive value, false-positive value, false-negative value, and accuracy, were determined using comparison analysis against results from RT-PCR as the gold standard. The difference in sensitivity of the Q Ag test used with nasopharyngeal swab samples versus nasal cavity swab samples was assessed using Fisher’s exact test. The agreement of the Q Ag tests between the nasopharyngeal swab and the nasal cavity swab samples was assessed based on the kappa index. We also performed sensitivity analyses by restricting COVID-19-positive cases according to DSO (1–4, 5–7, and ≥8) and Ct values (Ct < 15, 15 ≤ Ct < 25, 25 ≤ Ct ≤ 35). When no symptoms occurred, DSO was defined as the days from sample collection to RT-PCR confirmation. The mean differences in the Ct values for the *E* and *RdRp* between negative and positive samples based on the Q Ag test were assessed using Student’s *t*-test. The correlation of the color intensity scale values from the Q Ag test with Ct values from RT-PCR was evaluated using Spearman correlation analysis. A *p*-value of <0.05 was considered statistically significant. We performed all statistical analyses using SAS software ver. 9.4 (SAS Institute Inc., Cary, NC, USA) and R version 3.6.3.

## 3. Results

### 3.1. Sensitivity of the Q Ag Test Using Nasopharyngeal Samples

For nasopharyngeal swab samples, the sensitivity of the Q Ag test was 81.7% (95% confidence interval, CI 72.7–90.7%) based on the RT-PCR results (Table 1). The specificity and positive predictive value were both 100%, and the negative predictive value was 88.9% (95% CI, 83.2–94.6%). Overall accuracy was 92.6% (95% CI, 88.7–96.5%).

### 3.2. Sensitivity of the Q Ag Test Using Nasal Cavity Samples

For the nasal cavity swab samples, the sensitivity of the Q Ag test was 77.5% (95% CI 67.8–87.2%) based on the RT-PCR results. The specificity and positive predictive value were both 100%, and the negative predictive value was 86.7% (95% CI, 80.6–92.8%). Overall accuracy was 90.9% (95% CI, 86.6–95.1%) (Table 1).

### 3.3. Agreement between Tests with the Nasal Cavity and Nasopharyngeal Swabs

The agreement of the Q Ag test results between the nasal cavity and nasopharyngeal swab samples is presented in Table 2. The Cohen’s kappa index value was 0.78 (95% CI, 0.60–0.96), indicating substantial agreement between the two types of specimens.

### 3.4. Sensitivity Analysis according to DSO

We performed a sensitivity analysis to assess whether the performance of the Q Ag test was affected by changes in certain conditions. First, we evaluated the sensitivity of the Q Ag test by restricting COVID-19-positive cases according to DSO. When DSO was between 1 and 4, the sensitivities of the Q Ag tests using nasopharyngeal and nasal cavity specimens were 96.4% (95% CI, 89.6–100.0%) and 89.3% (95% CI, 77.8–100.0%), respectively (Table 3). In restricted subjects with a DSO of 5–7, the sensitivity decreased (for nasopharyngeal: 77.4%, 95% CI, 62.7–92.1%; for nasal cavity: 74.2%, 95% CI, 58.8–89.6%). When the DSO was ≥8, the sensitivity was 58.3% (95% CI, 30.4–86.2%) for both nasopharyngeal and nasal cavity samples. No statistical difference in the sensitivity of the Q Ag test between nasopharyngeal swab samples and nasal cavity swab samples was found (*p* = 0.82). Nevertheless, our study showed a significant difference in the sensitivities of the samples that were taken at an extended number of days after symptom onset (≥8 days) compared to those taken earlier (1–4 days) (*p <* 0.01).

### 3.5. Sensitivity Analysis according to Ct Value

Furthermore, we performed sensitivity analysis by classifying COVID-19 positives according to Ct values of the *E* and *RdRp* (Table 4). In this analysis, we performed Q Ag testing using specimens diagnosed as positive by RT-PCR, within which Ct value intervals of *E* and *RdRp* were divided into three groups, i.e., Ct < 15, 15 ≤ Ct < 25, and 25 ≤ Ct ≤ 35. The sensitivity of the Q Ag test for samples with Ct values for the *E* < 15 was 100% for both nasopharyngeal and nasal cavity swab specimens. When we restricted positive cases to Ct values between 15 and 25, the sensitivities of the Q Ag test slightly declined to 95% (95% CI, 88.3–100%) and 87.5% (95% CI, 88.3–100%) for nasopharyngeal and nasal cavity specimens, respectively. However, the sensitivity of Q Ag test significantly decreased to 26.7% for both nasopharyngeal and nasal cavity specimens when we restricted the positive cases to Ct values of the *E* to 25 ≤ Ct ≤ 35. We also observed a similar pattern when we used the Ct values of *RdRp* to restrict COVID-19 cases (data not shown).

### 3.6. Comparison of Ct Values between Positive and Negative Samples Based on Q Ag Test

The mean (SD) Ct value for the *E* was 18.5 (5.2) for Q-Ag-positive specimens and 29.2 (4.0) for Q-Ag-negative nasopharyngeal specimens (*p* < 0.01) (Appendix A). The mean (SD) Ct for the *E* was 18.4 (5.3) for Q-Ag-positive specimens and 27.5 (5.3) for Q-Ag-negative nasal cavity specimens (*p* < 0.01). Spearman’s rank coefficient of correlation (rho) between Q Ag test color intensity and Ct value for the *E* was −0.876 for nasopharyngeal and −0.725 for the nasal cavity samples (both *p* < 0.01). These results suggest that Q Ag test color intensity and Ct value have a strong negative correlation. When the color intensity was stronger, the Ct value tended to be lower (Figure 2). Observation of the correlation of the Q Ag test and Ct values of the *RdRp* showed similar results as those for the *E* (data not shown).

## 4. Discussion

Before taking the first steps in this study, we had three questions. First, is the RAT useful for diagnosing COVID-19? Second, is the nasal cavity an optimal site for specimen collection for RAT? Third, does the color intensity of RAT correlate with viral load? To answer these questions, we evaluated the performance of the Q Ag test kit, a RAT that has been widely used in South Korea, using specimens that were taken from the nasopharynx and the nasal cavity. We also evaluated the correlation between the Ct values from RT-PCR and the RAT color intensity.

Our study demonstrated that the overall sensitivity of the Q Ag test exceeded 81% and 77% for nasopharyngeal swab and nasal cavity swab specimens, respectively, with a 100% specificity for both sampling sites. There was no significant difference in the sensitivity between these two locations (*p* = 0.82). These values of sensitivities are acceptable according to the US Food and Drug Administration (FDA) regulations, which mention that an antigen test should have an at least 80% sensitivity and 98% specificity [21].

The sensitivity and accuracy of RATs have become a topic of discussion that has triggered numerous studies. One systematic review reported that the overall sensitivity of RATs was 56.2% (95% CI 29.5–79.8%) [5]. Another comparative analytical study that was conducted in South Korea using the RATs that were also used in our study (SD Biosensor Q Ag) showed a sensitivity range of 60.0–71.4% based on the site of sample collection [22], and 17.5% for admitted patients [23]. These reported ranges of sensitivities are low compared to our results.

This discrepancy in RAT sensitivity may be due to several factors, including differences in the methods of collection and handling of specimens, DSO, type of assays, disease severity, or the type of sample used. Previous studies that reported poor performance of RATs used stored specimens rather than fresh specimens [6] or did not include specific information regarding DSO ranges [23]. Meanwhile, in our study, we used fresh swab specimens, and the DSO was correctly recorded. Although some studies [24,25] have reported that storage of samples did not affect RAT sensitivity, evidence that storage conditions affect antigen test sensitivity was recently published [26]. The sensitivity of RAT declined from 75.3% for fresh specimens to 70.9% for frozen specimens [26]. Therefore, we expect that accurate timing in the performance of the RAT (i.e., DSO) and the storage conditions of specimens highly affect RAT sensitivity. Hence, the results of our study, which demonstrated the relatively higher sensitivity of RATs compared to the studies mentioned above [5,22,23], are reasonable, as we used fresh specimens and precise timing for sample collection (within 7 DSO). Moreover, there have also been some studies reporting a relatively high performance of RAT for the detection of SARS-CoV-2. A study that was conducted in the Netherlands reported that the overall Q Ag test sensitivity was 84.9% in a non-hospitalized symptomatic population, which increased to 95.8% for specimens collected before 8 DSO [27]. From the same study, when the Ct value of the *E* was <25, the sensitivity of the RAT increased to 99.1% [27]. Their study showed a better sensitivity than ours and, most importantly, they included a large population (*n* = 970). A study from South Korea also reported that the sensitivity and specificity of Q Ag test reached 89.2% and 96.0%, respectively [28]. In addition, a study on specimens obtained by nasopharyngeal and oropharyngeal swab demonstrated a clinical sensitivity for RAT of up to 94.7% for specimens collected within a week of DSO compared with RT-PCR [29].

We noticed that the sensitivity of the Q Ag test was significantly lower in the specimens that were collected after an extended time (≥8 DSO) compared to the specimens that were collected within 5 DSO. This finding is in accordance with previously reported results from the Netherlands [27]. The viral load has been reported to reach its peak (108.1 copies) at 4.3 days after onset and is estimated to decline at rate of 0.17 log10 units per day [30]. Hence, our finding that demonstrated a higher sensitivity of the Q Ag test in the specimens that were collected before 5 DSO is plausible.

As been reported before, Ct values are inversely related to viral RNA copy number [31]. A lower Ct value implies a high accumulation of viral load. Therefore, we restricted our evaluation based on the Ct values of the *E* and *RdRp*. When the Ct values of both *E* and *RdRp* were less than 25, the sensitivity exceeded 90% for nasopharyngeal specimens and 85% for nasal cavity specimens. However, when the Ct values were >25, the sensitivity declined to as low as 26% and 23% for the *E* and *RdRp*, respectively. Nevertheless, the overall sensitivity of the Q Ag test in our study was still higher than previously reported data [22]. We assume that this discrepancy is due to the difference in the specimen composition that was used. As we mentioned above, when the Ct values were >25, we observed a significant decline in the sensitivity of the Q Ag test. Therefore, when the proportion of specimens with Ct values >25 was greater, the overall sensitivity was lower. We noticed that only a few patients (around 7%) had Ct values > 25 in our study, whereas there were 48.1% with these Ct values in a Korean nationwide surveillance study [22]. Therefore, as demonstrated in our study, the Q Ag test showed relatively good performance for diagnosis of COVID-19 during early DSO and when Ct values were <25.

Most studies about RAT sensitivity have used either nasopharyngeal or oropharyngeal specimens [22,23,27,28]. The specimen collection site is one of the most important factors that determine the sensitivity and accuracy of diagnosis. Adequate specimen collection in a virus-rich area with optimal technique is critical for the diagnosis of COVID-19. A nasopharyngeal swab is highly recommended for collecting specimens for SARS-CoV-2 testing [32,33]. However, a nasopharyngeal swab has its own disadvantages, such as the potential to cause discomfort, pain, and even bleeding. In addition, it is essential to have a specialist to collect specimens from the nasopharynx as they must wear personal protective equipment. Therefore, a specimen collection method that is easier and causes less pain is required.

Nasal swabs have been recommended as an alternative specimen collection method for SARS-CoV-2 detection in symptomatic patients [34]. Acquiring a specimen from the nasal cavity for diagnosis of COVID-19 is pain-free, compliant, and comfortable; moreover, self-collection is possible. A previous study demonstrated that the antigen test using the QuickNavi-COVID19 Ag kit with anterior nasal samples showed 72.5% sensitivity and 100% specificity [35].

In our study, we demonstrated an overall good performance of the Q Ag test in nasopharyngeal swab specimens (81.7%), and relatively good performance in nasal cavity swab specimens (77.5%). Given that the sensitivity of the Q Ag test when using a specimen from the nasal cavity was considerably comparable and acceptable for the detection of SARS-CoV-2, we conclude that a nasal cavity swab may be a good option as an alternative specimen collection method for detection of SARS-CoV-2 when nasopharyngeal sampling is hard to perform due to shortage of resources such as specialized medical personnel or protection equipment.

Viral load is a crucial factor in determining the sensitivity of RATs [3,6,29,36]. The sensitivity of the Q Ag test was significantly higher when the Ct values of the *E* and *RdRp* were <25. The evaluation of RAT was based on the color intensity that appeared on the cassette, where the color intensities depend on the amount of SARS-CoV-2 antigen present in the sample. The distributions of the Q Ag color intensity and Ct values showed a substantial correlation, especially for nasopharyngeal samples. This finding implies that the color intensity of the Q Ag test result corresponds with the viral load. When the viral loads are high (indicated by lower Ct values), the color intensities are stronger. The correlation between these two parameters is consistent with previously reported results [27], suggesting that the subjective interpretation of color intensity in rapid antigen testing could be used as a surrogate indicator to estimate viral load.

The wild-type of SARS-CoV-2 was dominantly detected in South Korea since its first reported case in January 2020 to early 2021. Given that this study was conducted on December 2020 to January 2021, only wild-type SARS-CoV-2 was used to evaluate performance of the Q Ag test. A recent published study reported that RAT showed high sensitivity to detect most of variant of concerns (VOCs), i.e., alpha, beta, and gamma as high as wild-type [37]. Therefore, we assumed that the Q Ag test that used in our study may give an akin result to test other variants of SARS-CoV-2. The differences in the sensitivity of Q Ag test may occur among variants of SARS-CoV-2 considering the mutations that arises, however Q Ag test may still give a comparable result for screening of SARS-CoV-2 as the first step of diagnosis of COVID-19.

Finally, the sensitivity of RAT may also be affected by the personnel who conduct the assessment. Even though RAT is relatively easy to perform, specimen collection especially the one from the nasopharyngeal cavity may become a great challenge for the inexperienced personnel. An inaccurate sampling procedure due to the lack of experience may result in false negative, which leads to a lower degree of sensitivity and specificity of the RAT itself. Therefore, well-trained healthcare personnel become one of the crucial factors that determines the performance of the RAT. Nonetheless, given that in our study, we also found that Q Ag test showed a good performance for detection of SARS-CoV-2 from specimens that were obtained from less difficult sampling sites, i.e., nasal cavity, we expect that the Q Ag test could be a handy point-of-care testing kit for detection of SARS-CoV-19 that can be used on daily basis when well-trained healthcare personnel are limited.

## 5. Conclusions

In conclusion, we confirmed that a relatively high sensitivity was obtained for the Q Ag test when fresh specimens of DSO ≤ 5 or Ct < 25 were used. RAT could be used as a screening tool in particular situations, such as highly suspicious contacts or triage in an emergency department. Moreover, considering that the Q Ag assay provided 100% specificity in our study as well as in a Korean nationwide surveillance study [22], this RAT may be a favorable specific detection method for SARS-CoV-2. A high-specificity test will correctly rule out almost everyone who does not have the disease and will not generate many false-positive results. False positives should be avoided to prevent unnecessary additional testing and inappropriate isolation measures.

Furthermore, since the results obtained from the nasal cavity showed comparable sensitivity to those from nasopharyngeal specimens, the nasal cavity could be carefully considered an alternative site for sample collection for the diagnosis of COVID-19.

## Figures and Tables

**Figure 1 diagnostics-12-00710-f001:**
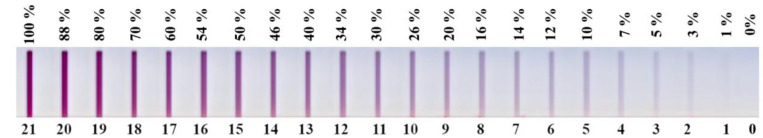
Color intensity scale of the STANDARD Q COVID-19 Ag test (Q Ag) (SD Biosensor, Suwon, Korea). The color intensities that appeared on the test kits were compared with the color scale and were recorded within the range of 0 to 21 as negative to very strong, respectively.

**Figure 2 diagnostics-12-00710-f002:**
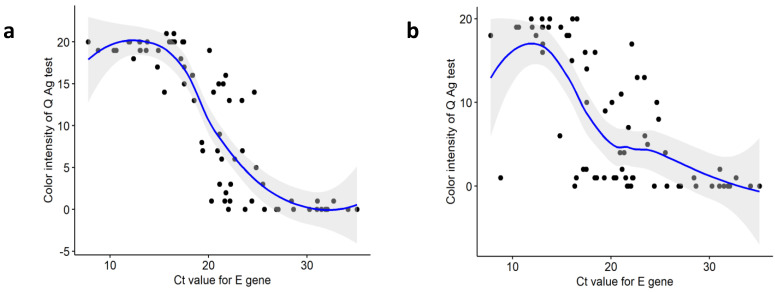
Spearman’s correlation analysis of the color intensity of the rapid antigen test (0: negative; 1: very weak; 21: very strong) and Ct values for the *E* determined by real-time PCR. (**a**) nasopharyngeal swab specimens (rho = −0.876, *p* < 0.01), (**b**) nasal cavity swab specimens (rho = −0.725, *p* < 0.01).

**Table 1 diagnostics-12-00710-t001:** Diagnostic performance of the STANDARD Q COVID-19 Ag test compared to real-time PCR.

Diagnostic Performance	Nasopharyngeal Specimen		Nasal Cavity Specimen	
	Value	95% CI	Value	95% CI
Sensitivity (%)	81.7	72.7–90.7	77.5	67.8–87.2
Specificity (%)	100.0	100.0–100.0	100.0	100.0–100.0
Positive predicted value (%)	100.0	100.0–100.0	100.0	100.0–100.0
Negative predicted value (%)	88.9	83.2–94.6	86.7	80.6–92.8
False positive value (%)	0.0	0.0–0.0	0.0	0.0–0.0
False negative value (%)	18.3	9.3–27.3	22.5	12.8–32.3
Accuracy (%)	92.6	88.7–96.5	90.9	86.6–95.1

Abbreviations: CI: confidence interval.

**Table 2 diagnostics-12-00710-t002:** Agreement of STANDARD Q COVID-19 Ag test between the nasal cavity and nasopharyngeal specimens.

Nasal Cavity Specimen	Nasopharyngeal Specimen		
	Positive	Negative	Total
Positive	54	1	55
Negative	4	12	16
Total	58	13	71

Cohen’s kappa index = 0.78 (95% CI, 0.60–0.96).

**Table 3 diagnostics-12-00710-t003:** Sensitivities of STANDARD Q COVID-19 Ag test according to time from symptom onset to sample collection (DSO).

		Nasopharyngeal Specimen		Nasal Cavity Specimen				
DSO	N	Sensitivity, %	95% CI	*p* *	N	Sensitivity, %	95% CI	*p* *
Overall	58	81.7	72.7–90.7		55	77.5	67.8–87.2	0.82
1–4 days	27	96.4	89.6–100.0		25	89.3	77.8–100.0	
5–7 days	24	77.4	62.7–92.1	0.11	23	74.2	58.8–89.6	0.12
≥8 days	7	58.3	30.4–86.2	<0.01	7	58.3	30.4–86.2	<0.01

* *p* for comparison between DSO of 5–7 days and ≥8 days vs. 1–4 days; Abbreviations: N, number of COVID-19 Ag positives, CI, confidence interval.

**Table 4 diagnostics-12-00710-t004:** Sensitivities of STANDARD Q COVID-19 Ag test according to Ct values for *E* and *RdRp*.

	Nasopharyngeal Specimen				Nasal Cavity Specimen			
Ct Value	N	Sensitivity, %	95% CI	*p* *	N	Sensitivity, %	95% CI	*p* *
Overall	58	81.7	72.7–90.7		55	77.5	67.8–87.2	0.82
** *E* **								
Ct < 15	16	100.0	100.0–100.0		16	100.0	100.0–100.0	
15 ≤ Ct < 25	38	95.0	88.3–100.0	<0.01	35	87.5	77.3–97.8	0.01
25 ≤ Ct ≤ 35	4	26.7	4.3–49.1	<0.01	4	26.7	4.3–49.1	<0.01
** *RdRp* **								
Ct < 15	22	100.0	100.0–100.0		21	95.5	86.8–100.0	
15 ≤ Ct < 25	33	91.7	82.6–100.0	0.06	31	86.1	74.8–97.4	0.07
25 ≤ Ct ≤ 35	3	23.1	0.2–46.0	<0.01	3	23.1	0.2–46.0	<0.01

* *p* for comparison between 15 ≤ Ct < 25 and 25 ≤ Ct ≤ 35 vs. Ct value <15; Abbreviations: N, number of COVID-19 Ag positives; CI, confidence interval.

## Data Availability

Data of this study can be available on reasonable request.

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
