# Peer review of "Evaluation of the Diagnostic Accuracy of Nasal Cavity and Nasopharyngeal Swab Specimens for SARS-CoV-2 Detection via Rapid Antigen Test According to Specimen Collection Timing and Viral Load"

_diagnostics, 2022, doi:10.3390/diagnostics12030710_

Round 1

Reviewer 1 Report

The study presented the diagnostic accuracy of nasal cavity and nasopharyngeal swab specimens for SARS-CoV-2 detection via RAT according to specimen collection timing and viral load. The sensitivity of the RAT for nasal cavity specimens exceeded 89% for <5 DSO and 86% for Ct of <25. Overall, the experiment of this work was well organized and the results should be interest of the readership of Diagnostics. 

However, the following points should be explained in the discussion.
1. What type of SARS was the sample tested in this study?
2. Does the detection sensitivity and accuracy differ depending on the type of SARS?

Author Response

The authors would like to thank the Reviewers for their specific and helpful comments on the manuscript. The authors have carefully taken the comments into consideration and have made a revision on the manuscript to address the Reviewers’ concerns. In this revision version, the authors have specified the variant of SARS. 

Responses to the Reviewer 1's commets:

  1. What type of SARS was the sample tested in this study?

Response: This study was conducted during late 2020 to January 2021. During this period, the wild type of SARS-CoV-2 (wild-type Wuhan-1) was the dominant type of SARS-CoV-2 that was detected in South Korea. Hence, the type of SARS-CoV-2 that tested in this study is wild type (wild-type Wuhan-1). (Page 8, lines 331-340).

  1. Does the detection sensitivity and accuracy differ depending on the type of SARS?

Response: In this study we only used wild-type of SARS-CoV-2 for evaluation the performance of RAT. Nevertheless, a recent published study reported that RAT showed high sensitivity to detect most of variant of concerns (VOCs), i.e. alpha, beta, and gamma as high as wild-type. Therefore, we assumed that the Q Ag test that used in our study may give an akin result to test other variants of SARS-CoV-2. The differences in the sensitivity of Q Ag test may occur among variants of SARS-CoV-2 considering the mutations that arises, however Q Ag test may still give a comparable result for screening of SARS-CoV-2 as the first step of diagnosis of COVID-19.

Reviewer 2 Report

In this paper, the authors present a fast diagnostic approach for SARS-Cov-2 detection through antigen test. Considering the soundness and significance of this result, I would recommend for immediate publication.

Author Response

The authors would like to thank the Reviewers for their specific and helpful comments on the manuscript. The authors have carefully taken the comments into consideration and have made a revision on the manuscript to address the Reviewers’ concerns.

Response to the Reviewer 2's comment:

In this paper, the authors present a fast diagnostic approach for SARS-Cov-2 detection through antigen test. Considering the soundness and significance of this result, I would recommend for immediate publication.

Response:

We appreciate the generous comments from the Reviewer on our manuscript. 

Reviewer 3 Report

Interesting and well written study regarding the Evaluation of the diagnostic accuracy of nasal cavity and naso-pharyngeal swab specimens for SARS-CoV-2 detection via rapid antigen test according to specimen collection timing and viral load. 

The study is interesting since it reports important conclusions regarding everyday life.

Some minor comments: 

  • in Introduction I would underline the importance of triage in this pandemic phase, and how this situation has modified our approach to patients, especially in emergency situations. You can find some details in the following papaer for example: Iezzi R et al. Longitudinal study of interventional radiology activity in a large metropolitan Italian tertiary care hospital: how the COVID-19 pandemic emergency has changed our activity. Eur Radiol. 2020 Dec;30(12):6940-6949. doi: 10.1007/s00330-020-07041-y. Epub 2020 Jun 30. PMID: 32607633; PMCID: PMC7326392.  Papi G et al. Unprotected stroke management in an undiagnosed case of Severe Acute Respiratory Syndrome Coronavirus 2 infection. J Stroke Cerebrovasc Dis. 2020 Sep;29(9):104981. doi: 10.1016/j.jstrokecerebrovasdis.2020.104981. Epub 2020 May 23. PMID: 32807416; PMCID: PMC7245230.
  • Materials & Methods: generally clear and concise. Statistical analysis is relatively simple but sufficient for the argument.
  • Results: nothing to concern
  • Discussion: I would underline the important role of the operator performing the test. I think that specificity and sensitivity could be largerly influenced by operator's experience. A poorly done test is more likely negative. Could you comment on that? Could you also comment on applying your conclusions in clinical practice and in every day life too?

Author Response

The authors would like to thank the Reviewers for their specific and helpful comments on the manuscript. The authors have carefully taken the comments into consideration and have made a revision on the manuscript to address the Reviewers’ concerns. In this revision version, the authors have  underlined the importance of triage in the pandemic phase, and added comments about the important role of the operator on the performance of rapid test kits according to the Reviewer’s comments.

Some minor comments: 

  1. in Introduction I would underline the importance of triage in this pandemic phase, and how this situation has modified our approach to patients, especially in emergency situations. You can find some details in the following paper for example: Iezzi R et al. Longitudinal study of interventional radiology activity in a large metropolitan Italian tertiary care hospital: how the COVID-19 pandemic emergency has changed our activity. Eur Radiol. 2020 Dec;30(12):6940-6949. doi: 10.1007/s00330-020-07041-y. Epub 2020 Jun 30. PMID: 32607633; PMCID: PMC7326392.  Papi G et al. Unprotected stroke management in an undiagnosed case of Severe Acute Respiratory Syndrome Coronavirus 2 infection. J Stroke Cerebrovasc Dis. 2020 Sep;29(9):104981. doi: 10.1016/j.jstrokecerebrovasdis.2020.104981. Epub 2020 May 23. PMID: 32807416; PMCID: PMC7245230.

Response: Early detection of SARS-CoV-2 is crucial in the triage of patient with acute/subacute diseases during COVID-19 pandemic. Therefore, application of RAT as a rapid detection platform may give an early and rapid diagnosis of COVID-19, especially for asymptomatic patients with high-risk acute disease. Early diagnosis of COVID-19 help giving the best clinical judgement before delivering healthcare, and at the same time minimize cross-infection of COVID-19 among patients, protect the healthcare workers from possibility of infection and maintain the service in the healthcare center (Page 2, lines 58-64).

  1. Materials & Methods: generally clear and concise. Statistical analysis is relatively simple but sufficient for the argument.

Response: We appreciate the thoughtful comment from the Reviewer on our manuscript. 

  1. Results: nothing to concern

Response: We thank the Reviewer for the positive comment.   

  1. Discussion: I would underline the important role of the operator performing the test. I think that specificity and sensitivity could be largerly influenced by operator's experience. A poorly done test is more likely negative. Could you comment on that? Could you also comment on applying your conclusions in clinical practice and in every day life too?

Response:

We agree with your idea. In our opinion, the sensitivity of RAT may also be affected by the personnel who conduct assessment. Even though, RAT is relatively easy to perform, specimen collection especially the one from nasopharyngeal cavity may become a great challenge for the inexperienced operator. An inaccurate sampling procedure due to the lack experience may result in false negative, which lead into lower degree of sensitivity and specificity of the RAT itself. Therefore, well-trained healthcare personnel become one of crucial factor that determine the performance of RAT.

Nonetheless, given that in our study we also found that Q Ag test is showed a good performance for detection of SARS-CoV-2 from specimens that obtained from less difficult sampling site i.e., nasal cavity, we expect that Q Ag test is a handy point of care testing kit for detection of SARS-CoV-19 that can be used in daily basis when a well-trained healthcare personnel are limited.  (Page 9, Lines 341-351). 

Round 2

Reviewer 1 Report

The authors properly explained my comments in this revision.

So, I recommend accept in present form.